# Pervaporation of Aqueous Ethanol Solutions through Rigid Composite Polyvinyl-Alcohol/Bacterial Cellulose Membranes

**Tănase Dobre** [1], **Claudia Ana Maria Patrichi** [1,*], **Oana Cristina Pârvulescu** [1,*] **and Ali A. Abbas Aljanabi** [2]

[1] Chemical and Biochemical Engineering Department, University POLITEHNICA of Bucharest, 1-6 Gheorghe Polizu, 011061 Bucharest, Romania; tghdobre@gmail.com
[2] Al Mussaib Technical College, Al-Furat Al-Awsat Technical University, Babylon 54003, Iraq; dr.ali.aljanabi@atu.edu.iq
*  Correspondence: patrichi.claudia@gmail.com (C.A.M.P.); oana.parvulescu@upb.ro (O.C.P.)

**Abstract:** The paper focuses on synthesis, characterization and testing in ethanol-water separation by pervaporation of new membrane types based on polyvinyl alcohol (PVA) and bacterial cellulose (BC). A technology for obtaining these membranes deposited on a ceramic support is presented in the experimental section. Three PVA-BC composite membranes with different BC content were obtained and characterized by FTIR, SEM and optic microscopy. The effects of operating temperature (40–60 °C), permeate pressure (18.7–37.3 kPa) and feed ethanol concentration (24–72%wt) on total permeate flow rate (0.09–0.23 kg/m$^2$/h) and water/ethanol selectivity (5–23) were studied based on an appropriate experimental plan for each PVA-BC membrane. Statistical models linking the process factors to pervaporation performances were obtained by processing the experimental data. Ethanol concentration of the processed mixture had the highest influence on permeate flow rate, an increase in ethanol concentration leading to a decrease in the permeate flow rate. All 3 process factors and their interactions had positive effects on membrane selectivity. Polynomial regression models were used to assess the effect of BC content in the dried membrane on pervaporation performances. Values of process performances obtained in this study indicate that these membranes could be effective for ethanol-water separation by pervaporation.

**Keywords:** pervaporation; PVA-bacterial cellulose membrane; FTIR analysis; membrane selectivity; permeate flow rate

## 1. Introduction

The most attractive point of the pervaporation process is the separation of the organic solvents [1–3], especially the water removal. It presents a high level of performance and it is economically efficient. In the field of solvents dehydration, there have been many cases in which pervaporation technology has been employed and successfully implemented on an industrial scale [4,5]. Currently, more than 90 industrial units are in operation world-wide for the dehydration of ethanol, isopropanol, ethyl acetate and multipurpose solvents [6].

There are many studies focused on the development of new membranes that can be used for the pervaporation process as well as for other processes involving membranes. For the pervaporation process, the focus is on the enhancement of membranes, in order to be more efficient, with longer-term stability and life, and on the economic factor, expressed in terms of the accepted cost. Although many types of materials have been used in the development of membranes, the composite membranes based on polyvinyl-alcohol (PVA) are the ones mostly used in the pervaporation process [7–11]. Due to its numerous hydroxyl groups (Figure 1), PVA exhibits a high water permeability. Moreover, dried PVA-based membranes have better abrasion resistance, mechanical properties and selectivity for hydrophilic species [12]. The high degree of swelling of PVA determines an increase in the free volume of its network and a decrease in the chemical interactions with the permeating species, resulting in low membrane selectivity and stability [12].

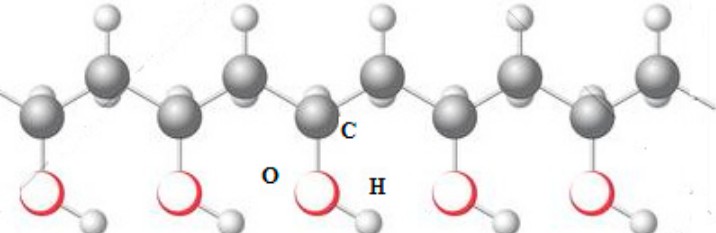

**Figure 1.** Polyvinyl alcohol (PVA) structure.

Various techniques have been employed to obtain enhanced performances of PVA membranes [13–16]. Praptowidodo [13] studied the influence of the cross-linking degree on PVA-based membrane performances by adding glutaraldehyde (GA) as a cross-linking agent. An increase in the amount of GA led to a decrease in the degree of swelling, resulting in a decrease in the permeate flow rate and an improvement of membrane selectivity. The mechanical properties of composite membranes based on PVA can be highly improved by adding suitable reinforcement materials in PVA. Huang et al. [17] prepared a composite hydrogel containing PVA and polyvinyl pyrrolidone (PVP) with wrapped multiwalled carbon nanotubes. Unlike a PVA-based membrane, this composite membrane exhibited about a 133% increase in the tensile strength and 63% in the tear strength.

Bacterial cellulose (BC) is a special cellulose with a nanoscale structure, produced by selected aerobic Gram-negative bacteria, which are active on saccharide substrates at a certain composition, pH and temperature [18]. BC is a very pure product, with a high degree of crystallinity and polymerization and with an interesting mechanical behavior [19,20]. BC exhibits superior structural properties compared to plant cellulose [21]. It has been proven that BC, either pure or composite, can be used as a valuable biopolymer in various fields (e.g., industrial, medical, food processing) [22–24]. The hydrophilic nature of pure BC and BC-based composites confers these types of membranes selectivity in water transport, if a binary mixture containing water and another compound passes through the membrane. During the BC membrane pervaporation process of binary ethanol-water mixture, as the water content is high, it results in a high permeate flow rate and a low selectivity.

Recently, much research has been focused on PVA-BC composites, especially because both components present strong hydrophilic behavior. In PVA-BC composites, BC appears as fiber, film or bulk material. The aim is to obtain a membrane with good mechanical properties, high chemical stability as well as good selectivity and permeability for transferable species [13]. For PVA-BC nanocomposite layers, Gea et al. [25] presented different preparation methods; the results showed that the presence of PVA led to a reduction in Young's modulus and in an increase in toughness compared to a pure BC layer. Tan et al. [26] and Chiciudean et al. [27,28] used a layered PVA-BC composite hydrogel, which exhibited excellent mechanical properties.

High permeate flow rates and acceptable water selectivities were obtained for pervaporation of ethanol-water mixtures through BC membranes [29]. However, due to membrane cracking at drying and also excessive swelling, a high pervaporation surface is not possible. Pervaporation performances of BC-PVA membranes, obtained by impregnating BC gel with PVA and cross-linking with GA [30], were superior to those of BC membranes. However, these types of membranes have the same disadvantage of cracking and swelling, which does not allow the development of pervaporation surfaces required by applications with higher working capacity.

In this paper, we assess PVA-BC membranes, with BC distributed discreetly and controlled in PVA, applied on supports of any shape and size, and processed by cross-linking with total water removal, cover all practical requirements of a pervaporation process with water transport. Accordingly, hydrophilic fibers of BC were mixed with PVA and GA and the mixture was deposited on a ceramic support, to obtain rigid PVA-BC membranes with a high operational effectiveness when used for water separation from various mixtures. The main aim of the paper was the study of pervaporation performances

during the dehydration of ethanol-water solutions using rigid PVA-BC composites deposited on a ceramic support.

## 2. Materials and Methods

### 2.1. Materials

The pervaporation rigid membranes were prepared from PVA powder type $M_w$ 89,000–98,000 (Sigma Aldrich), BC gel ($0.995$ g/cm$^3$) having 96% water, fibrils with diameters less than 50 nm, a crystallinity index of $78-90\%$ and a polymerization degree of $2500-4000$ (UPB Mass Transfer Laboratory) and GA (Merck) as a cross-linking agent. The membranes were deposited on a high porosity ceramic tube support (type TC-00314, Research, Design and Production Center for Refractive (CCPPR), Alba Iulia, Romania).

### 2.2. Membrane Preparation

There were two steps involved in preparing the disperse systems used to obtain PVA-BC membranes. The first step involved the preparation of a PVA solution having a PVA content of 75 g/L (by dissolving 12 g of PVA in 150 mL of distilled water, under moderate agitation, at 60 °C. In the second step, BC gel was ground by using a disintegrator knife and then mixed with the PVA solution to obtain a PVA/BC ratio of 2 (12 g PVA and 6 g BC), 1 (12 g PVA and 12 g BC) and 0.66 (12 g PVA and 18 g BC), respectively. Thus, three PVA-BC disperse systems were obtained, the concentrations of PVA and BC being 7.14% and 3.57% for the first system (S1), 6.90% and 6.90% for the second system (S2), 6.67% and 10% for the third system (S3), respectively. The disperse systems were stored at 5 °C before being used.

Rigid PVA-BC membranes were obtained as follows: (i) the membrane support (ceramic tube) was cleaned with distilled water, dried at 105 °C and then weighed; (ii) 30 g of disperse system (S1, S2 or S3) was mixed with 1.2 g of GA, which was used as a cross-linking agent; (iii) the membrane mixture was applied on the ceramic tube with a brush; after the first layer was applied, the system was dried at 60 °C for 15–20 min and then weighed; the second layer was applied and the procedure was repeated until the thin layer reached the desired mass (1.3–1.8 g on the ceramic support with an external diameter of 0.035 m and a length of 0.25 m); (iv) supported membrane was dried at 90 °C for 90–120 min, then weighed to determine the final mass and thickness of the active membrane layer. Depending on the disperse system (S1–S3) from which they were prepared, the membranes were coded as follows: PVA-BC 12/6 (M1), PVA-BC 12/12 (M2) and PVA-BC 12/18 (M3).

Figure 2 shows the stages of preparation of rigid PVA-BC membranes applied on the ceramic tube. The computed thickness of membranes was about 45 μm, according to some tests performed with a PosiTector 6000 apparatus (DeFelsko inspection Instruments). In order to have membrane samples for physicochemical analysis (FTIR, SEM), film samples were applied on a glass plate.

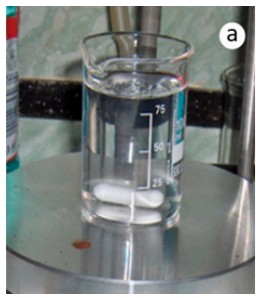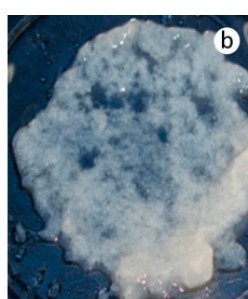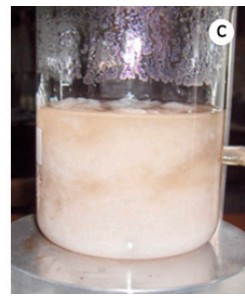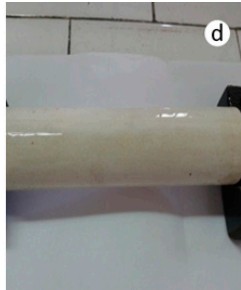

**Figure 2.** The stages of preparation of rigid polyvinyl alcohol-bacterial cellulose (PVA-BC) membrane on ceramic support: (**a**) PVA solution, (**b**) wet ground BC, (**c**) PVA-BC-glutaraldehyde (GA) mixture, (**d**) membrane applied on support.

### 2.3. Pervaporation Tests

Pervaporation experiments involving these rigid membranes were carried out in a laboratory equipment (Figure 3). The membrane (1b) deposited on the ceramic support (1c) and the glass tube (1a) of the pervaporation unit (1) were heated by an electric resistance heater, the heating eliminating the permeate condensation on the glass tube. The pervaporation chamber was connected to the vacuum and to the systems for measuring/control the process temperature and pressure as well as the permeate flow rate.

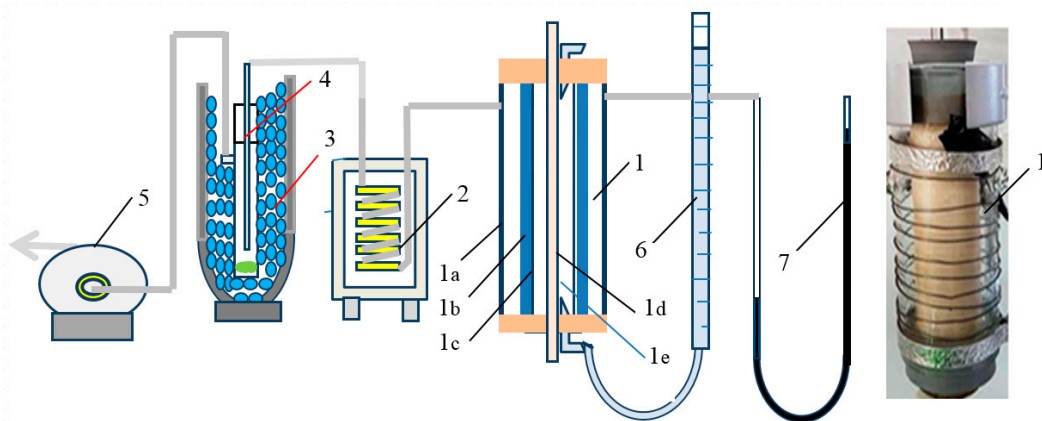

**Figure 3.** Laboratory equipment for pervaporation studies with rigid membranes deposited on the external surface of a ceramic tube: (**1**) pervaporation unit, (**1a**) glass tube, (**1b**) PVA-BC membrane, (**1c**) ceramic support, (**1d**) tube for heating the processed liquid, (**1e**) ethanol-water mixture, (**2**) electric cooler, (**3**) Dewar vessel with crushed ice, (**4**) steam condenser with central tube, (**5**) vacuum pump, (**6**) precision burette for measuring the permeate flow rate, (**7**) mercury manometer.

As shown in Figure 3, the ethanol-water mixture (1e) was placed in the precision burette (6), which was connected to the liquid pervaporation chamber (1). The liquid mixture inside the pervaporation unit passed through the membrane, where the water was preponderantly removed and the ethanol was concentrated in the pervaporation system. On the external side of the membrane, the water and the associated ethanol were vaporized due to the low pressure on the membrane surface. The vapor phase was condensed in a cooling system, which consists of two parts: the first part is a glass coil refrigerator working at a temperature of 5–6 °C, whereas the second part is a cold trap containing ice. The purpose of using this cooling system is to provide as much cold space as possible for the condensation of all permeating vapor.

Pervaporation tests were performed at different levels of temperature (40, 50 and 60 °C), pressure (140, 210 and 280 mm Hg, i.e., 18.7, 28.0 and 37.3 kPa) and ethanol concentrations in the feed solution (24, 48 and 72%wt). The ethanol concentrations in the feed and in the condensed permeate were determined by measuring the refractive index.

Characteristic working procedure of a pervaporation test was as follows: (i) the ceramic tube of pervaporation unit was filled with ethanol-water mixture and the working liquid level in the burette was measured; (ii) working temperature and pressure levels were selected and the vacuum pump was started; (iii) the time ($\Delta\tau$) after which the liquid level in the burette decreased by V = 2 mL was measured; (iv) *the burette was refilled and other 3 measurements of* $\Delta\tau$ *were performed*; (v) the refractive index of the condensate collected in the steam condenser was measured.

### 2.4. Process Parameters

Total permeate flow rate, $N_A$, was determined using Equation (1), where A is the effective membrane area, $\rho_f$ the density of feed solution and $\Delta\tau$ the time after which the liquid level in the burette decreased by V = 2 mL.

$$N_A = \frac{V\rho_f}{A\Delta\tau} \tag{1}$$

Ethanol concentrations in the feed and permeate samples were estimated using an Atago Abbe refractometer (Atago, Japan). Based on the experimental data obtained for the refractive index at 25 °C and refractive index-concentration curves [31], the mass fractions of ethanol were obtained. Water/ethanol selectivity, $\alpha_{w/et}$, was calculated using Equation (2), where $X$ and $Y$ represent the mass fractions of species in the feed and permeate.

$$\alpha_{w/et} \;=\; \frac{\frac{Y_w}{Y_{et}}}{\frac{X_w}{X_{et}}} \;=\; \frac{\frac{1-Y_{et}}{Y_{et}}}{\frac{1-X_{et}}{X_{et}}} \tag{2}$$

## 3. Results and Discussions

Pervaporation data corresponding to 4 experimental runs (measurements of $\Delta\tau$) using PVA-BC 12/6 membrane (M1) are summarized in Table 1, where $c_{et}$ is the concentration of ethanol in the feed solution, $t$ the temperature, $p$ the pressure, $V$ the volume of feed solution passed through membrane in the time $\Delta\tau$, $n_f$ and $n_p$ are the refractive indexes for the feed solution and permeate.

**Table 1.** Experimental data for PVA-BC 12/6 membrane (M1).

| $c_{et}$ (%wt) | $n_f$ | $t$ (°C) | $p$ (mm Hg) | $V$ (mL) | $\Delta\tau$ in Run | | | | $n_p$ |
|---|---|---|---|---|---|---|---|---|---|
| | | | | | I | II | III | IV | |
| 48 | 1.3592 | 50 | 210 | 2 | 66 | 63 | 66 | 72 | 1.3376 |
| 24 | 1.3485 | 60 | 280 | 2 | 61 | 48 | 45 | 45 | 1.3345 |
| 24 | 1.3485 | 60 | 140 | 2 | 45 | 35 | 17 | 10 | 1.3335 |

Characteristic FTIR spectra of PVA-BC 12/6, PVA-BC 12/12 and PVA-BC 12/18 membranes used in the pervaporation experiments are shown in Figure 4. Due to the fact that the FTIR spectra only analyze the vibration of the molecular groups, it is not expected to have many differences among the spectra of membranes, which differ only in BC content. For a deeper analysis, the FTIR spectra for BC, PVA and PVA cross-linked with GA are provided (Figures 5 and 6).

It is known that characteristic peaks of PVA are revealed at 3263 cm$^{-1}$ (O-H stretching), 2921 cm$^{-1}$ (C-H stretching), 1413 cm$^{-1}$ (CH$_2$ bending), 1235 cm$^{-1}$ (OH plane bending), 1085 cm$^{-1}$ (C-O stretching), 916 cm$^{-1}$ (CH$_2$ rocking) and 828 cm$^{-1}$ (C-C stretching). For PVA cross-linked with GA (Figure 6), almost all peaks are present (with small displacements, due to the structure stiffening). Moreover, with small displacements, all these peaks appear for PVA-BC membranes (Figure 4).

According to Figures 5 and 6, BC and PVA cross-linked with GA have 5 peaks almost similar, i.e., those at 3343 and 3267 cm$^{-1}$ (O-H stretching), 2895 and 2912 cm$^{-1}$ (C-H stretching), 1314 and 1327 cm$^{-1}$ (OH bending, alcohol), 1162 and 1141 cm$^{-1}$ (C-O stretching, ester or ether), 1054 and 1087 cm$^{-1}$ (C-O stretching, alcohol). Also, a peak at 1629 cm$^{-1}$ (C-OH absorbed water), can be identified in the FTIR spectrum of BC (Figure 5) and similar peaks, but of much lower intensity, appear in characteristic FTIR spectra of PVA-BC membranes (Figure 4), i.e., at 1640.0 cm$^{-1}$ for PVA-BC 12/6 and 1658.7 cm$^{-1}$ for PVA-BC 12/18. The low intensity of these peaks, probably due to the heat treatment (60−90 min at 90 °C) to complete cross-linking, may indicate that these membranes do not trend to retain water and swell.

The spectra of composite membranes (M1-M3) shown in Figure 4 indicate a slight displacement of characteristic peaks of BC and PVA cross-linked with GA as well as the appearance of new peaks, specific to these membranes. The results plotted in Figure 4 highlight 7 peaks for PVA-BC 12/6 membrane (M1), 9 peaks for PVA-BC 12/12 membrane (M2) and 11 peaks for PVA-BC 12/18 membrane (M3). Accordingly, new peaks appear as the concentration of BC in the membranes increases. This suggests that each PVA-BC membrane has its own structure, which depends on BC content. It is expected that the

behavior of the membranes in the pervaporation process will depend on the BC content in the membrane, which will be considered as a process factor in the future.

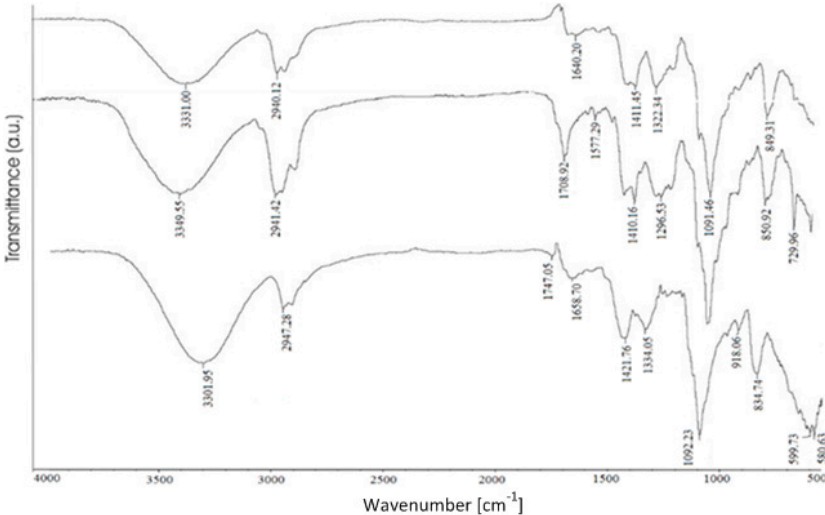

**Figure 4.** FTIR spectra for PVA-BC 12/6 (**top**), PVA-BC 12/12 (**middle**) and PVA-BC 12/18 (**bellow**) composite membranes after preparing and drying at 90 °C.

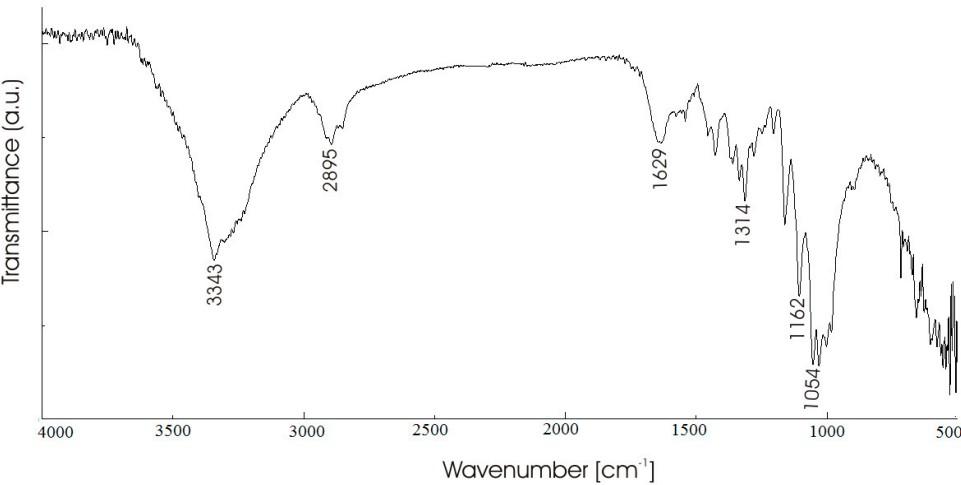

**Figure 5.** FTIR spectrum of BC used for composite membranes synthesis.

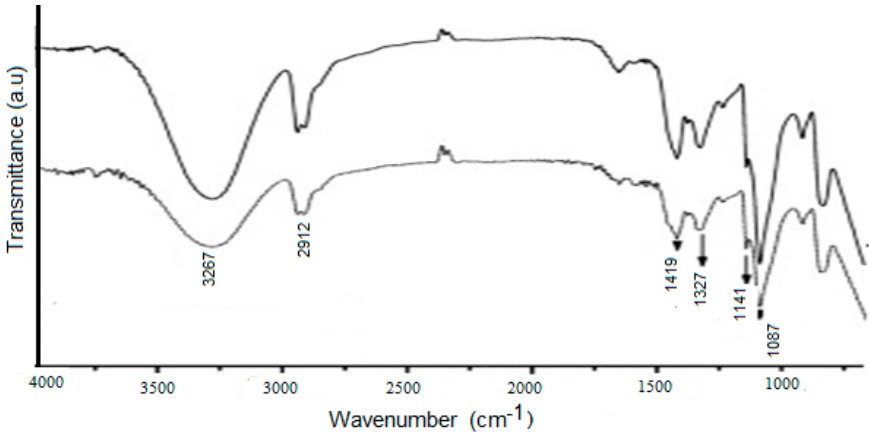

**Figure 6.** FTIR spectra for PVA (**top**) and PVA cross-linked with GA (**bellow**).

Optic microscope images of BC fibrils in PVA solutions are shown in Figure 7. We observed the BC distribution as micro agglomerates with concentrations consistent with BC content in the mixture.

SEM images presented in Figure 8 show composites membrane in which the compact PVA film is furrowed by BC fibers having diameters less than 2 μm. It appears that the density of fibrils in the films is proportional to the BC content of the disperse system used to prepare the membranes.

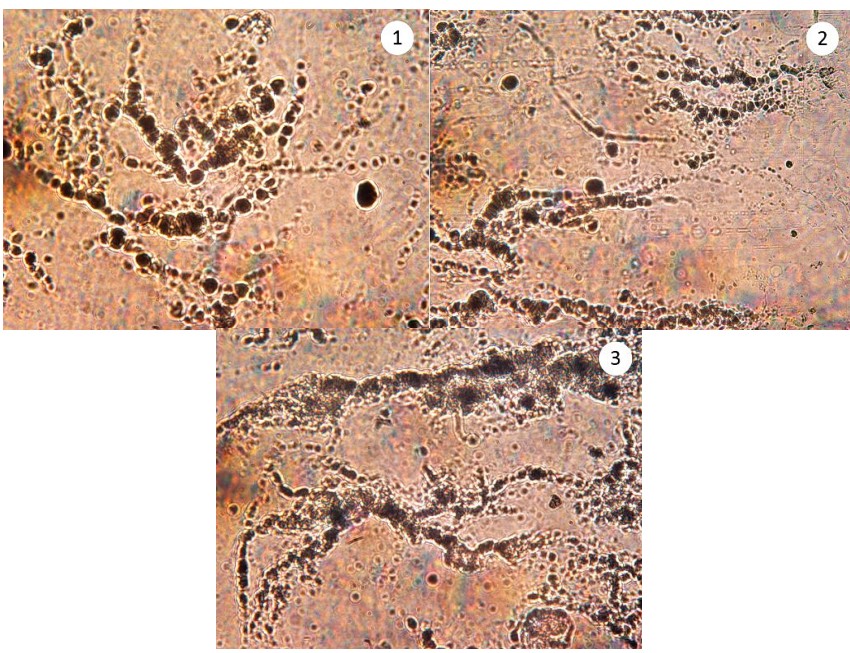

**Figure 7.** Optic microscope images (×10) of PVA-BC 12/6 (**1**), PVA-BC 12/12 (**2**) and PVA-BC 12/18 (**3**) mixtures before the start of membrane cross-linking.

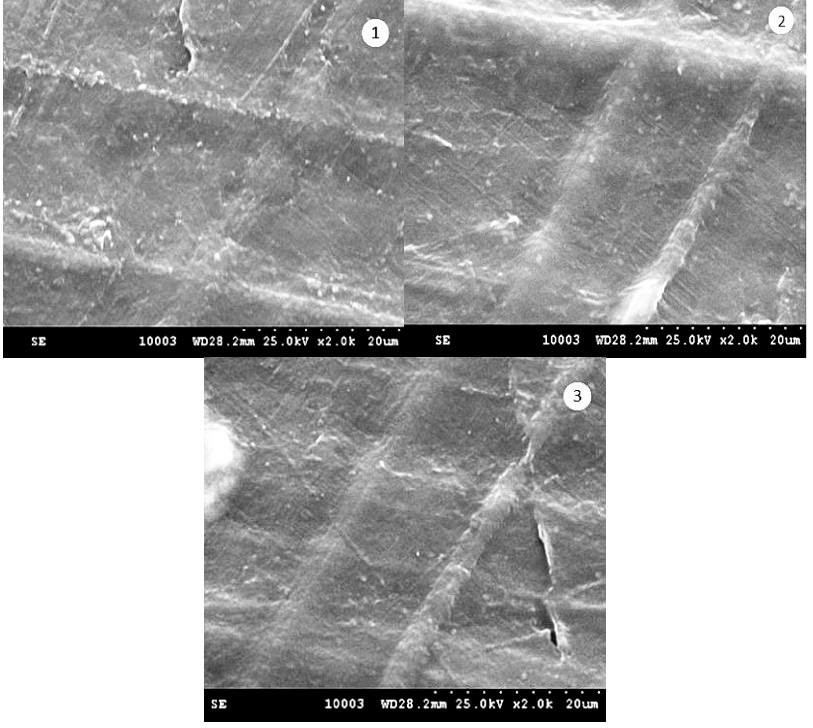

**Figure 8.** SEM images of PVA-BC 12/6 (**1**), PVA-BC 12/12 (**2**) and PVA-BC 12/18 (**3**) membranes.

Pervaporation tests for each membrane used in the experimental study were organized according to a factorial plan with 2 levels for each factor, i.e., pervaporation temperature ($t$), pervaporation pressure ($p$) and ethanol concentration in the feed solution ($c_{et}$). Process response variables in terms of permeate flow rate and membrane selectivity depending on process factors for PVA-BC 12/12 membrane (M2) are summarized in Table 2. The dimensionless factors ($x_1$, $x_2$ and $x_3$) were calculated using Equations (3)–(5), where the $c$ subscript indicates the center of experimental plan. Characteristic experimental data matrices of $2^3$ factorial experiments for PVA-BC 12/6 (M1), PVA-BC 12/12 (M2) and PVA-BC 12/18 (M3) membranes, containing the values of process responses at various levels of dimensionless factors, are given in Tables S1–S3. Levels of process response variables for all pervaporation measurements, i.e., 0.09–0.23 kg/m$^2$/h for total permeate flow rate and 5–23 for water/ethanol selectivity, are summarized in Figure 9. These levels are in the same ranges as those characterizing the cellulose-silica membranes [21,32] or some special cellulose membranes [33,34].

The results presented in Figure 9 highlight the following issues:

(i) pervaporation flow rate increased with an increase in $t$ and $p$ as well as it decreased with an increase in $c_{et}$ for all M1–M3 membranes, the effect of $c_{et}$ being higher [for $c_{et}$ = 24%wt (runs 1-4), the values of pervaporation flow rate (0.14–0.23 kg/m$^2$/h) were almost double than those corresponding to $c_{et}$ = 72%wt (runs 5-8), i.e., 0.09–0.11 kg/m$^2$/h]; in all experimental runs 1-4 ($c_{et}$ = 24%wt), the pervaporation flow rate was higher for M1 membrane (having the lowest content of BC) and lower for M3 membrane (having the highest content of BC), whereas in all experimental runs 5-8 ($c_{et}$ = 72%wt), it was almost constant for M1–M3 membranes;

(ii) for M3 membrane, water/ethanol selectivity increased with an increase in $t$, $p$ and $c_{et}$;

(iii) for M1 and M2 membranes, the values of selectivity in runs 3 and 4 ($c_{et}$ = 24%wt and $p$ = 37.3 kPa), i.e., 4.5–9.7, were lower than those in runs 7 and 8 ($c_{et}$ = 72%wt and $p$ = 37.3 kPa), i.e., 14.1–18.4; characteristic selectivities of M1 in runs 3 and 4 were almost double than those of M2, whereas selectivities of M1 in runs 7 and 8 were slightly lower (up to 5%) than those of M2; except for runs 5 and 6 ($c_{et}$ = 72%wt and $p$ = 18.7 kPa), water/ethanol selectivities increased with an increase in $t$ for M1 and M2 membranes;

(iv) M1-M3 membranes had similar selectivities in runs 1 and 2 ($c_{et}$ = 24%wt and $p$ = 18.7 kPa), i.e., 7.5–9.5, whereas in runs 3-8 the selectivities of M3 (9.5–22.9) were higher than those of M1 (6.8–17.9) and M2 (4.5–18.42).

**Table 2.** Process factors and responses for PVA-BC 12/12 membrane (M2).

| No. | $t$ (°C) | $x_1$ | $p$ (kPa) | $x_2$ | $c_{et}$ (%wt) | $x_3$ | $N_A$ (kg/m$^2$/h) | $c_{Etp}$ (%wt) | $\alpha_{w/et}$ |
|---|---|---|---|---|---|---|---|---|---|
| 1 | 40 | −1 | 18.7 | −1 | 24 | −1 | 0.182 | 3.18 | 8.6 |
| 2 | 60 | 1 | 18.7 | −1 | 24 | −1 | 0.203 | 2.92 | 9.5 |
| 3 | 40 | −1 | 37.3 | 1 | 24 | −1 | 0.195 | 5.24 | 4.7 |
| 4 | 60 | 1 | 37.3 | 1 | 24 | −1 | 0.218 | 4.92 | 5.1 |
| 5 | 40 | −1 | 18.7 | −1 | 72 | 1 | 0.089 | 20.63 | 8.9 |
| 6 | 60 | 1 | 18.7 | −1 | 72 | 1 | 0.097 | 24.71 | 6.8 |
| 7 | 40 | −1 | 37.3 | 1 | 72 | 1 | 0.092 | 13.95 | 14.8 |
| 8 | 60 | 1 | 37.3 | 1 | 72 | 1 | 0.112 | 11.50 | 18.8 |
| 9 | 50 | 0 | 28 | 0 | 48 | 0 | 0.151 | 8.30 | 9.2 |
| 10 | 50 | 0 | 28 | 0 | 48 | 0 | 0.148 | 8.05 | 9.6 |
| 11 | 50 | 0 | 28 | 0 | 48 | 0 | 0.147 | 7.81 | 9.9 |

$$x_1 = \frac{t - t_c}{\Delta t}, \quad t_c = 50\,°C, \quad \Delta t = 10\,°C \tag{3}$$

$$x_2 = \frac{p - p_c}{\Delta p}, \quad p_c = 28\,kPa, \quad \Delta p = 9.3\,kPa \tag{4}$$

$$x_3 = \frac{c_{et} - c_{et,c}}{\Delta c_{et}}, \quad c_{et,c} = 48\%\text{wt}, \quad \Delta c_{et} = 24\%\text{wt} \tag{5}$$

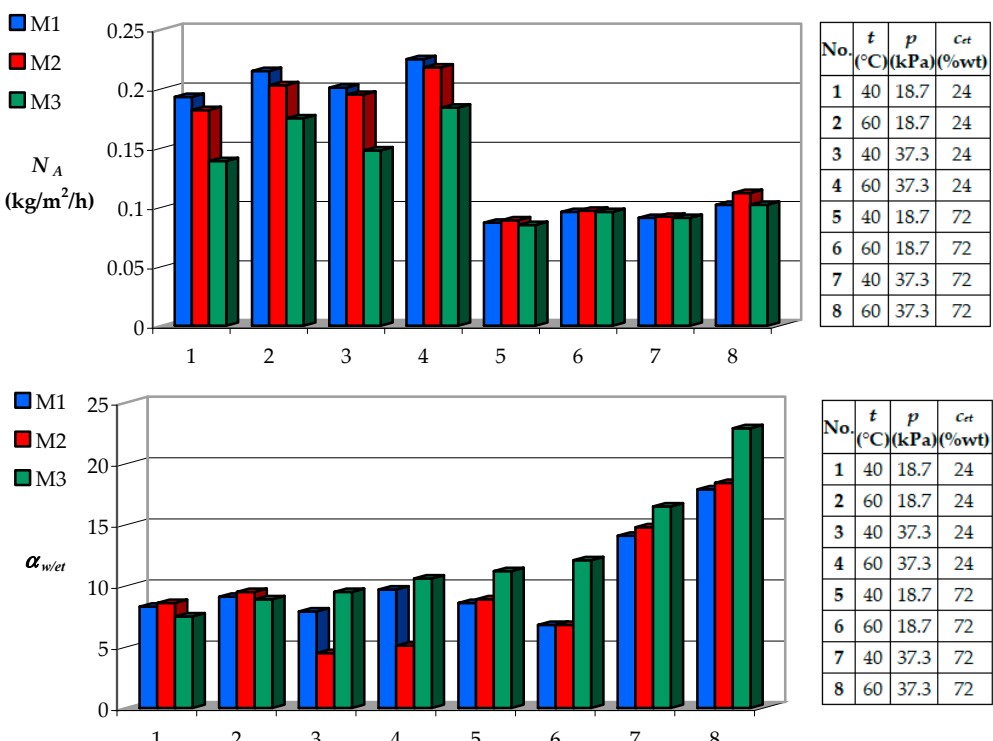

**Figure 9.** Permeate flow rate and membrane selectivity for pervaporation through PVA-BC membranes (M1: PVA-BC 12/6, M2: PVA-BC 12/12, M3: PVA-BC 12/18).

Data presented in Figure 9 were used to quantify the effect of pervaporation factors on response variables. Experimental data for each membrane ($M_k$, $k$ = 1..3) were processed as follows: (i) characteristic regression analysis of a $2^3$ experimental plan was used to obtain values of significant regression coefficients in dependences $N_{Ak}(x_1,x_2,x_3)$ and $\alpha_{w/etk}(x_1,x_2,x_3)$ given by Equations (6) and (7) [35]; (ii) significant regression coefficients in Equations (6) and (7), $\beta_{ik}$ and $\alpha_{jk}$, were processed resulting in 2 polynomial regression models, $\beta_i(x_4) = b_{0i} + b_{1i}x_4 + b_{2i}x_4^2$ and $\alpha_j(x_4) = a_{0j} + a_{1j}x_4 + a_{2j}x_4^2$, where $x_4$ represents BC concentration in the dried membrane; analytical expressions $N_A(x_1,x_2,x_3,x_4)$ and $\alpha_{w/et}(x_1,x_2,x_3,x_4)$ were determined based on these polynomial regression models.

$$N_{Ak}(x_1,x_2,x_3) = \beta_{0k} + \beta_{1k}x_1 + \beta_{2k}x_2 + \beta_{3k}x_3 + \beta_{12k}x_1x_2 + \beta_{13k}x_1x_3 + \beta_{23k}x_2x_3 + \beta_{123k}x_1x_2x_3 \tag{6}$$

$$\alpha_{w/etk}(x_1,x_2,x_3) = \alpha_{0k} + \alpha_{1k}x_1 + \alpha_{2k}x_2 + \alpha_{3k}x_3 + \alpha_{12k}x_1x_2 + \alpha_{13k}x_1x_3 + \alpha_{23k}x_2x_3 + \alpha_{123k}x_1x_2x_3 \tag{7}$$

Tables S1−S3 contain values of significant regression coefficients in Equations (6) and (7) for the membranes used in the pervaporation experiments. The significance of regression coefficients was assessed based on 3 center-point runs performed for each membrane (Table 2 and Table S1–S3) according to the procedure detailed in our previous papers [36,37]. Taking into account these significant coefficients, Equations (6) and (7), which express the dependences $N_{Ak}(x_1,x_2,x_3)$ and $\alpha_{w/etk}(x_1,x_2,x_3)$, turn into Equations (8) and (9) for PVA-BC 12/6 membrane (M1), Equations (10) and (11) for PVA-BC 12/12 membrane (M2), Equations (12) and (13) for PVA-BC 12/18 membrane (M3). Data presented in Tables S1–S3 indicate that the experimental values of process performances and those calculated using Equations (8)–(13) are in a good agreement.

$$N_{A1}(x_1,x_2,x_3) = 0.151 + 0.008x_1 - 0.057x_3 \tag{8}$$

$$\alpha_{w/et1}(x_1,x_2,x_3) = 10.3 + 2.1x_2 + 1.55x_3 + 0.825x_1x_2 + 2.05x_2x_3 \tag{9}$$

$$N_{A2}(x_1, x_2, x_3) = 0.149 + 0.009x_1 + 0.0057x_2 - 0.051x_3 \tag{10}$$

$$\alpha_{w/et2}(x_1, x_2, x_3) = 9.58 + 1.1x_2 + 2.65x_3 + 0.68x_1x_2 + 3.25x_2x_3 + 0.75x_1x_2x_3 \tag{11}$$

$$N_{A3}(x_1, x_2, x_3) = 0.127 + 0.012x_1 - 0.034x_3 - 0.0062x_1x_3 \tag{12}$$

$$\alpha_{w/et3}(x_1, x_2, x_3) = 12.4 + 1.225x_1 + 2.475x_2 + 3.275x_3 + 1.55x_2x_3 \tag{13}$$

Equations (8)–(13) highlight the following issues:

(i) $N_{A1}$, $N_{A2}$ and $N_{A3}$, which are expressed by Equations (8), (10) and (12), depend linearly on temperature ($x_1$) and ethanol concentration in the feed mixture ($x_3$) and $x_3$ has a higher effect; an increase in $x_1$ and a decrease in $x_3$ lead to an increase in $N_{Ak}$ ($k$ = 1..3); for PVA-BC 12/12 membrane (M2), there is a low linear influence of the pressure from the pervaporation unit [the term $+ 0.0057x_2$ in Equation (10)]; for PVA-BC 12/18 membrane (M3), the interaction between temperature and composition of processed mixture diminishes the permeate flow rate [the term $- 0.0062x_1x_3$ in Equation (12)];

(ii) $\alpha_{w/et1}$, $\alpha_{w/et2}$ and $\alpha_{w/et3}$, which are expressed by Equations (9), (11) and (13), depend on process factors and their interactions; all regression coefficients are positive, consequently, an increase in $x_1$, $x_2$ and $x_3$ leads to an increase in $\alpha_{w/etk}$ ($k$ = 1..3).

These findings are similar to those obtained for pervaporation of ethanol-water system or isopropanol-water mixture through BC [29], BC-PVA [30] or cellulose based membranes [38].

The regression coefficients in Equations (8)–(13) depend on the membrane type, which is related to BC concentration in the dried membrane ($c_{BC}$ = 2.4–7.2%wt). The values of regression coefficients in Equations (8), (10) and (12), $\beta_{ik}$ ($i$ = 0, 1, 2, 3, 13, $k$ = 1..3), and Equations (9), (11) and (13), $\alpha_{jk}$ ($j$ = 0, 1, 2, 3, 12, 23, 123, $k$ = 1..3), are summarized in Tables 3 and 4, where dimensionless BC concentration ($x_4$) is defined by Equation (14). Polynomial regression models, i.e., $\beta_i(x_4) = b_{0i} + b_{1i}x_4 + b_{2i}x_4^2$ and $\alpha_j(x_4) = a_{0j} + a_{1j}x_4 + a_{2j}x_4^2$, obtained by processing the values of $\beta_{ik}$ and $\alpha_{jk}$ ($k$ = 1..3) are also given in Tables 3 and 4. Taking into account these polynomial regression models, Equations (8), (10) and (12) turn into Equation (15) and Equations (9), (11) and (13) into Equation (16), which express the permeate flow rate ($N_A$) and membrane efficiency ($\alpha_{w/et}$) depending on 4 process factors, i.e., temperature ($x_1$), pressure ($x_2$), ethanol concentration in the feed solution ($x_3$) and BC concentration in the dried membrane ($x_4$).

$$x_4 = \frac{c_{BC} - c_{BC,c}}{\Delta c_{BC}}, \quad c_{BC,c} = 4.8\%\text{wt}, \quad \Delta c_{BC} = 2.4\%\text{wt} \tag{14}$$

**Table 3.** Regression coefficients $\beta_{ik}$ ($i$ = 0, 1, 2, 3, 13, $k$ = 1..3) depending on BC content in the membrane.

| Parameter | Membrane Type | | | Polynomial Regression Model $\beta_i(x_4) = b_{0i} + b_{1i}x_4 + b_{2i}x_4^2$ |
|---|---|---|---|---|
| | **PVA-BC 12/6** ($k$ = 1) | **PVA-BC 12/12** ($k$ = 2) | **PVA-BC 12/18** ($k$ = 3) | |
| $c_{BC}$ (%wt) | 2.4 | 4.8 | 7.2 | - |
| $x_4$ | −1 | 0 | 1 | - |
| $\beta_{0k}$ | 0.151 | 0.149 | 0.127 | $0.149 - 0.012x_4 - 0.014x_4^2$ |
| $\beta_{1k}$ | 0.00825 | 0.009 | 0.012 | $0.009 + 0.056x_4 + 0.055x_4^2$ |
| $\beta_{2k}$ | 0 | 0.00575 | 0 | $0.00575 - 0.00575x_4^2$ |
| $\beta_{3k}$ | −0.057 | −0.051 | −0.034 | $-0.051 + 0.012x_4 + 0.0055x_4^2$ |
| $\beta_{13k}$ | 0 | 0 | 0.00625 | $0.0031x_4 + 0.0031x_4^2$ |

**Table 4.** Regression coefficients $\alpha_{jk}$ ($j = 0, 1, 2, 3, 12, 23, 123, k = 1..3$) depending on BC content in the membrane.

| Parameter | Membrane type | | | Polynomial Regression Model $\alpha_j(x_4) = a_{0j} + a_{1j}x_4 + a_{2j}x_4^2$ |
|---|---|---|---|---|
| | **PVA-BC 12/6** **($k = 1$)** | **PVA-BC 12/12 ($k = 2$)** | **PVA-BC 12/18 ($k = 3$)** | |
| $c_{BC}$ (%wt) | 2.4 | 4.8 | 7.2 | - |
| $x_4$ | −1 | 0 | 1 | - |
| $\alpha_{0k}$ | 10.3 | 9.58 | 12.4 | $9.58 + 1.05x_4 + 1.77x_4^2$ |
| $\alpha_{1k}$ | 0 | 0 | 1.225 | $0.613x_4 + 0.613x_4^2$ |
| $\alpha_{2k}$ | 2.1 | 1.27 | 2.475 | $1.127 + 0.188x_4 + 1.161x_4^2$ |
| $\alpha_{3k}$ | 1.55 | 2.65 | 3.275 | $2.65 + 0.862x_4 - 0.238x_4^2$ |
| $\alpha_{12k}$ | 0.825 | 0.68 | 0.05 | $0.68 - 0.388x_4 - 0.243x_4^2$ |
| $\alpha_{23k}$ | 2.05 | 3.25 | 1.55 | $3.25 - 0.25x_4 - 1.45x_4^2$ |
| $\alpha_{123k}$ | 0 | 0.75 | 0 | $0.75 - 0.75x_4^2$ |

$$N_A(x_1, x_2, x_3, x_4) = \left(0.149 - 0.012x_4 - 0.014x_4^2\right) + \left(0.009 + 0.056x_4 + 0.055x_4^2\right)x_1 + \left(0.00575 - 0.00575x_4^2\right)x_2 + \left(-0.051 + 0.012x_4 + 0.0055x_4^2\right)x_3 + \left(0.0031x_4 + 0.0031x_4^2\right)x_1x_3 \tag{15}$$

$$\alpha_{w/et}(x_1, x_2, x_3, x_4) = \left(9.58 + 1.05x_4 + 1.77x_4^2\right) + \left(0.613x_4 + 0.613x_4^2\right)x_1 + \left(1.127 + 0.188x_4 + 1.161x_4^2\right)x_2 + \left(2.65 + 0.862x_4 - 0.238x_4^2\right)x_3 + \left(0.68 - 0.388x_4 - 0.243x_4^2\right)x_1x_2 + \left(3.25 - 0.25x_4 - 1.45x_4^2\right)x_2x_3 + \left(0.75 - 0.75x_4^2\right)x_1x_2x_3 \tag{16}$$

In order to validate the statistical models described by Equations (15) and (16), 5 experimental runs were performed at $x_1 = x_2 = x_3 = x_4 = 0.5$, i.e., $t = 55\ °C$, $p = 32.7$ kPa, $c_{et} = 60$%wt and $c_{BC} = 6$%wt (PVA-BC 12/15 membrane). Experimental values of process performances for these replicates ($n = 5$), $N_{A,ex}$ and $\alpha_{w/et,ex}$, and those predicted by Equations (15) and (16), $N_{A,pr}$ and $\alpha_{w/et,pr}$, are summarized in Table 5. A test statistic for one sample $T$-test, denoted $T$ and defined by Equation (17), was calculated for each process performances ($P$) to assess whether the difference between the mean of experimental values and predicted value ($P_{ex.mn} - P_{pr}$) is statistically significant. Values of $P_{ex.mn}$, standard deviation ($SD$), standard error of the mean ($SE$), coefficient of variance ($CV = 100SD/P_{ex.mn}$), degrees of freedom ($df = n - 1$) and $T$ are specified in Table 5. The difference $P_{ex.mn} - P_{pr}$ is statistically non-significant for each process performance [the values of $T$ (1.147 and 1.236) are lower than its critical value for two tails at a significance level of 0.05 and $df = 4$, i.e., $T_{cr}(0.05,4) = 2.776$], which demonstrates the model validity. Accordingly, both Equations (15) and (16) can be used to design and simulate pervaporation units equipped with supported PVA-BC membranes, at levels of process factors in the ranges selected in the experimental study.

$$T = \frac{P_{ex,mn} - P_{pr}}{\frac{SD}{\sqrt{n}}} = \frac{P_{ex,mn} - P_{pr}}{SE} \tag{17}$$

**Table 5.** Experimental and calculated data for replicates used to validate the statistical models described by Equations (15) and (16).

| Run | $t$ (°C) | $p$ (kPa) | $c_{et}$ (%wt) | $c_{BC}$ (%wt) | $N_A$ (kg/m²/h) | | $\alpha_{w/et}$ | |
|---|---|---|---|---|---|---|---|---|
| | | | | | Experimental (*ex*) | Predicted (*pr*) by Equation (15) | Experimental (*ex*) | Predicted (*pr*) by Equation (16) |
| 1 | | | | | 0.153 | | 13.9 | |
| 2 | | | | | 0.139 | | 15.1 | |
| 3 | 55 | 32.7 | 60 | 6 | 0.146 | 0.1458 | 14.9 | 13.911 |
| 4 | | | | | 0.161 | | 13.4 | |
| 5 | | | | | 0.151 | | 14.2 | |
| $P_{ex,mn}$ | | | | | 0.150 | kg/m²/h | 14.3 | - |
| $SD$ | | | | | 0.0082 | kg/m²/h | 0.7036 | - |
| $SE$ | | | | | 0.0037 | kg/m²/h | 0.3146 | - |
| $CV$ | | | | | 5.46 | % | 4.92 | % |
| $df$ | | | | | 4 | | 4 | |
| $T$ | | | | | 1.147 | | 1.236 | |

## 4. Conclusions

The composite PVA-BC membranes deposited on ceramic supports can be very effective to remove water from solvents. Preparation, characterization and testing of rigid PVA-BC membranes applied on a ceramic support are presented in this paper. Three membranes types, i.e., M1, M2 and M3, with different concentration of BC in the dried membrane (2.4, 4.8 and 7.2%wt) were prepared and then tested in pervaporation experiments. An experimental equipment at a laboratory pilot scale was used to study the pervaporation of ethanol-water mixtures through supported PVA-BC membranes.

A two-level experimental plan based on 3 factors, i.e., temperature ($x_1$), permeation pressure ($x_2$) and ethanol concentration in the feed solution ($x_3$), was used in the experimental investigation of the pervaporation behavior of each membrane type. Regression equations linking these 3 factors to process performances in terms of permeate flow rate ($N_{Ak}$) and membrane selectivity ($\alpha_{w/etk}$) were obtained according to characteristic procedure of a $2^3$ factorial experiment for each membrane type ($M_k$, $k = 1..3$). Regression coefficients in these equations, $\beta_{ik}$ ($i = 0, 1, 2, 3, 13$) and $\alpha_{jk}$ ($j = 0, 1, 2, 3, 12, 23, 123$), were processed resulting in 2 polynomial regression models, $\beta_i(x_4) = b_{0i} + b_{1i}x_4 + b_{2i}x_4{}^2$ and $\alpha_j(x_4) = a_{0j} + a_{1j}x_4 + a_{2j}x_4{}^2$, where $x_4$ represents BC concentration in the dried membrane. Analytical expressions $N_A(x_1,x_2,x_3,x_4)$ and $\alpha_{w/et}(x_1,x_2,x_3,x_4)$ were determined based on these polynomial regression models. These relationships can be used to predict the pervaporation performances at levels of process factors in the ranges selected in the experimental study.

**Supplementary Materials:** The following are available online at https://www.mdpi.com/2227-9717/9/3/437/s1, Table S1: Data and results of regression analysis ($2^3$ experimental plan) for ethanol-water pervaporation through PVA-BC 12/6 membrane (M1), Table S2: Data and results of regression analysis ($2^3$ experimental plan) for ethanol-water pervaporation through PVA-BC 12/12 membrane (M2), Table S3: Data and results of regression analysis ($2^3$ experimental plan) for ethanol-water pervaporation through PVA-BC 12/18 membrane (M3).

**Author Contributions:** Methodology, investigation, formal analysis, writing-original draft preparation, T.D., C.A.M.P., O.C.P. and A.A.A.; writing-review and editing, T.D., O.C.P. and C.A.M.P. All authors have read and agreed to the published version of the manuscript.

**Funding:** This research received no external funding.

**Institutional Review Board Statement:** Not applicable.

**Informed Consent Statement:** Not applicable.

**Data Availability Statement:** Data are contained within the article.

**Conflicts of Interest:** The authors declare no conflict of interest.

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
