# Peer review of "Pervaporation of Aqueous Ethanol Solutions through Rigid Composite Polyvinyl-Alcohol/Bacterial Cellulose Membranes"

_processes, doi:10.3390/pr9030437_

Round 1

Reviewer 1 Report

The comments are included in the attached file.

Reviewer 2 Report

This work investigates the pervaporation performance of PVA-BC membranes by varying the membrane composition and process conditions. Generally, the paper is structured in a way that is hard to follow. There is little discussion on the effect of the different composition and conditions on the pervaporation performance. More characterisations will be required to understand the effects. My comments are provided below:

1) What is the main difference between this work and the literature? It appears that PVA-BC membranes are not new.  The use of PVA-BC for pervaporation is also not new. The purpose of this work is not clear. Please improve the introduction to clearly show the motivation of this work.

2) line 114: Why did the authors use a brush to prepare the membranes? How do you ensure the thickness of the membranes are consistent? Please provide the cross-sectional view of the tubular membranes (M1-3) to show the thickness.

3) All the results and discussion are all under "2. Materials and Methods." Please restructure the paper as the current version is hard to follow. Line 186: Is this the start of the Results and discussion?

4) The resolution of Fig. 6 is poor. 

5) More characterisation is needed to understand how the crosslinkers affect the performance. E.g., crosslinking degree, hydrophilicity etc.

6) Table 2 only provide the results for M2. Results for M1 and M3 are missing.

7) Line 235 to 333: These are hard to read. The details should all be moved to SI or methods. The actual discussion of the results is not available. The texts are description of how the statistical models are performed rather than the discussion about the membranes.

8) Fig. 9 is hard to follow. Instead, a figure on the effect of each parameter (concentration, temperature, pressure etc.) on the flux and selectivity (double y plot) may be more suitable.

Reviewer 3 Report

First of all, I would like to recommend the work of Zh. Mao et al. "Dehydration of isopropanol – water mixtures using a novel cellulose membrane prepared from cellulose / N-methylmorpholine-N-oxide / H2O solution". 2010. Separation and Purification Technology. 72 (1): 28-33
DOI: 10.1016 / j.seppur.2010.01.002. In this paper, very good transport and selectivity properties are achieved.

The introduction and conclusions of manuscript need to be revised. In the first case, conduct a deeper literary analysis from which to highlight the main advantages of using BC. Conclusions, on the contrary, can be simplified.

Lines 20, 21. It is better to remove this part from Abstract "were studied based on a 23 experimental plan for each PVA-BC membrane".
Line 54. Figure 1. Recommendation. Reduce the PVA formula to 2 C atoms per chain. Example: Fig. 1 - https://www.researchgate.net/publication323808560_Section_C_Physical_Sciences_DSC_and_TGA_Properties_of_PVA_Films_Filled_with_Na_2_S_2_O_3_5H_2_O_Salt or remove brackets and n.
Lines 73, 74. "The hydrophilic nature of pure BC and BC-based composites makes these types of membranes less permeable to organic species than to water." I disagree with the authors as it all depends on the method of membrane preparation and experimental conditions.
Lines 86-90. It is desirable to rewrite the purpose of the work. "hard" should be replaced by "rigid".
Line 94. Why didn't the authors "clean up" BC. After all, as you know, a large number of bacterial residues and others remain in BC (Makarov, IS; Shambilova, GK; Vinogradov, MI; Zatonskih, PV; Gromovykh, TI; Lutsenko, SV; Arkharova, NA; Kulichikhin, VG Films of Bacterial Cellulose Prepared from Solutions in N-Methylmorpholine-N-Oxide: Structure and Properties. Processes 2020, 8, 171. https://doi.org/10.3390/pr8020171) which can affect the further properties of the resulting membranes.

Information about the degree of polymerization BC will not be superfluous.
Lines 94, 103 etc. Delete "crude".
Lines 145-154. I think that such a detailed description of the method is superfluous.
Line 184. Remove "Some".

Lines 211 - 218. It is better to plot the spectra in one figure.

Lines 229-232. Poor quality optical microscopy and SEM photographs. They need to be replaced. It is also desirable to describe in more detail the observed difference for the samples.

Lines 294, 295. The authors claim that "regression coefficients in Equations (8) - (13) depend on membrane structure", however, little attention is paid to the study of the structure itself.

Round 2

Reviewer 1 Report

The revised version of the manuscript can be accepted for publication.

Author Response

Thank you very much for your time, kindness, and patience.

Reviewer 2 Report

thanks for revising the manuscript

Author Response

(The authors gave the same response as above.)

Reviewer 3 Report

Lines 119, 121 ... "solutions" should be replaced by "systems".
Lines 113-122. What viscosity did the S3 system have, as I know already for 1% BC in water, the viscosity of suspensions is 103-104 Pa * s (Vinogradov MI et al. Rheological Properties of Aqueous Dispersions of Bacterial Cellulose. Processes 2020, 8, 423. https: // doi.org/10.3390/pr8040423)?
Line 148. "were heated by an electric resistance" I suggest adding "heater"
Line 268. Check dimension kg / m2 / h. 

Author Response

This manuscript is a resubmission of an earlier submission. The following is a list of the peer review reports and author responses from that submission.

Round 1

Reviewer 1 Report

This work investigates the water-ethanol pervaporation performance of newly developed PVA-biocellulose membranes by varying the feed concentration, temperature and pressure. Extensive experiments have been performed but the presentation of results are a little challenging to read. My specific comments are provided below:

1) The difference between 'first', 'second', 'third' membranes are not clear under 2.3 membrane preparation.

2) It is suggested to name the 3 membranes rather than referring to them as first, second, or third. For example, the membrane can be named after their difference, PVA-BC-2:1

3) Figure 8: Scale bar is missing

4) Figure 9: Scale bar is hard to read

5) What is the thickness of the membranes?

6) Table 2 only contain results from 2nd membrane. For completeness, all the results should be provided (but in the Supporting Information)

7) Figure 10 is difficult to read as the experimental case 1 to 8 has no meaning. They are not variables. Instead, results can be plotted by changing 1 variable at a time. For example, at fixed pressure and concentration, how does the temperature affect the flux and selectivity? This could be plotted as a double y plot. Alternatively, a 3d plot can be used (for examples, with two variables as x and y and flux as z).

8) Discussion in page 14 is difficult to read. Do all 3 membranes show the same changes with concentration, temperature and pressure? If not, how do the composition of the membranes affect the flux and selectivity?

Reviewer 2 Report

The comments are included in the attached Word file.

Round 2

Reviewer 2 Report

The revised version of the manuscript does not solve the main problems I found. The developed model includes complex equations (19 parameters without physical meaning) and has not been tested with experimental results apart from the ones employed to the fitting of the equations. Therefore, I maintain my previous verdict and I consider the manuscript should be rejected.